# Prevalence of SARS-CoV-2 infection and immunity in a New York county in 2022 reveals frequent asymptomatic or undiagnosed infections

Casey L. Cazer[1,2]*, Jeanne W. Lawless[2], Parshad Mehta[2], Bettina Wagner[3], Diego G. Diel[3], Katherine R. McLaughlin[4], Jeffrey W. Bethel[5], Elizabeth F. Plocharczyk[6], Kevin J. Cummings[2], Genevive R. Meredith[2], Samantha Hillson[7], Robert Lawlis[8], Lara Parrilla[2], Benjamin D. Dalziel[9,10]

1 Department of Clinical Sciences, College of Veterinary Medicine, Cornell University, Ithaca, NY, 2 Department of Public and Ecosystem Health, College of Veterinary Medicine, Cornell University, Ithaca, NY, 3 Department of Population Medicine and Diagnostic Sciences, College of Veterinary Medicine, Cornell University, Ithaca, NewYork, United States of America, 4 Department of Statistics, College of Science, Oregon State University, Corvallis, Oregon, United States of America, 5 College of Public Health and Human Sciences, Oregon State University, Corvallis, Oregon, United States of America, 6 Cayuga Medical Center, Ithaca, NewYork, United States of America, 7 Tompkins County Health Department, Ithaca, NewYork, United States of America, 8 Cayuga Health Partners, Ithaca, NewYork, United States of America, 9 Department of Integrative Biology, College of Science, Oregon State University, Corvallis, Oregon, United States of America, 10 Department of Mathematics, College of Science, Oregon State University, Corvallis, Oregon, United States of America

* clc248@cornell.edu

## Abstract

Accurate and timely surveillance of SARS-CoV-2 prevalence and immunity is critical to local and national COVID-19 pandemic responses. Representative surveillance surveys reveal more accurate estimates of COVID-19 infection than other measures based on reported test results. Our main research objectives were (i) to provide local health department officials with prevalence estimates calculated from a representative sample to better inform their decision-making efforts in response to the COVID-19 pandemic and (ii) to identify characteristics associated with COVID-19 infections among high-risk groups. Three municipalities were sampled at one time-point (February, April, or October 2022) using a 2-stage cluster sampling design. Participants provided anterior nares swabs, which were tested for SARS-CoV-2 with a RT-PCR and for nucleocapsid protein and receptor binding domain antibodies by multiplex Luminex assay. Participants completed a survey on socio-demographics, SARS-CoV-2 prevention behaviors and attitudes, and vaccination and infection history. A total of 233 individuals from 221 households provided anterior nares swabs, and 215 samples were linked to survey data. After adjusting for study design, the household prevalence of PCR-positive tests was less than 5%, but approximately half of the population had antibodies from a prior infection and most (81% to 92%) had antibodies from either infection or vaccination. Discrepancies between self-reported

**Data availability statement:** Data and code to reproduce the analysis are archived (https://zenodo.org/records/14514382) and are available with an appropriate IRB protocol to protect against identification of human subjects. Please contact the corresponding author (clc248@cornell.edu) and Cornell University IRB (irbhp@cornell.edu) to initiate an IRB protocol for data sharing.

**Funding:** This work was supported by a grant (#2021-72608) from the David and Lucile Packard Foundation. The funders had no role in study design, data collection and analysis, decision to publish, or preparation of the manuscript.

**Competing interests:** The authors have declared that no competing interests exist.

positive test and vaccination status and antibody results suggested a high prevalence of asymptomatic infection and waning antibody titers. County-level infection prevalences, estimated from the county test reporting system, were 16.6% in February, 19.1% in April, and 23.8% in October, substantially lower than the prevalence of individuals with antibodies from infection in the surveys, also supporting a high prevalence of asymptomatic or unconfirmed infections. The overall small sample size precluded an analysis of characteristics associated with active or past infection. In conclusion, surveillance surveys can provide timely data on infection status and immunity to support public health responses.

## Introduction

Disease surveillance is a cornerstone of public health practice, particularly to understand and control the spread of infectious pathogens [1]. Early in the COVID-19 pandemic, public health practitioners relied primarily on test positivity (i.e., the proportion of positive tests) to monitor the spread of SARS-CoV-2 [2–4]. These approximations result in disease incidence and prevalence estimates that are biased. The bias arises from multiple factors: there may be unreported or undiagnosed cases due to unequal test availability and access, the presence of asymptomatic infections, at-home testing, and the underlying prevalence of infection may vary among different populations being tested (e.g., test positivity in a clinic where patients are primarily symptomatic may be higher than test positivity in a random sample with lower prevalence and a high proportion of asymptomatic infection) [3–6]. The known risk of underestimating cases resulted in a call for public health officials to incorporate random sampling in COVID-19 tracking [7] and the development of a mathematical model to adjust test positivity rates for asymptomatic, undetected infections [8]. Surveys adopting representative sampling methods reveal much higher and more accurate estimates of COVID-19 infection than test positivity reporting [3,4,9–15].

Serologic surveillance and testing enable distinctions to be made between individuals who have been exposed to the virus (i.e., infected individuals) and those who have been vaccinated [3,5,12,16]. Assessing samples for antibodies specific to SARS-CoV-2 nucleocapsid protein (NP) improves the reliability of population prevalence rates of natural infection [3]. The presence of antibodies specific to SARS-CoV-2 receptor binding domains (RBD) in the absence of NP antibodies identifies individuals previously vaccinated who haven't been exposed to the virus [3,5,12,16]. Testing for hybrid immunity (i.e., immunity derived from both previous infection and vaccination) enables public health professionals to make timely recommendations to at-risk groups who have a low prevalence of hybrid immunity [16].

Finally, surveillance for behavioral factors that mitigate or promote COVID-19 infection risk is important in developing public health recommendations targeting groups at high risk [17–20], including in university settings [21]. Integrating surveillance for active infection with surveillance for preventive behaviors can be successful in managing COVID-19 infection in some settings [22].

This study aimed to provide local public health officials with timely, accurate estimates of SARS-CoV-2 infection, immunity from both past infections and vaccinations, and behaviors and attitudes related to SARS-CoV-2 spread in a rural New York State county by conducting representative surveillance surveys to help inform their pandemic decision-making responses.

## Methods

### Survey design

A multistage sampling design was used to identify a random sample of housing units and participants. A housing unit (HU) is a home, apartment, or other group of rooms that are occupied by an individual or group of people. In the first stage, ArcGIS was used to identify United States Census blocks in a municipality. Data on the number of households and population size was obtained from the 2020 Census using the *tidycensus* package for R (version 4.2.3), implemented in R Studio (version 2023.06.1). Blocks with a population of zero or with greater than 90% of the total population in group housing were excluded. The remaining blocks were merged manually, in an iterative process using Microsoft Excel, to form larger clusters that had at least 50 HU per cluster. Clusters were weighted by the number of HUs, and a random sample of clusters was selected using the *sample* function in R without replacement. Each cluster was previewed by one of the authors (CC) or a senior field staff member to identify safety and access concerns, for example, "No Soliciting" signs. If a large proportion of HUs in the cluster could not be accessed, for example, if an apartment complex did not allow soliciting, then the cluster was replaced with another randomly selected cluster.

In the second stage, field staff systematically selected HUs within each cluster by approaching HUs at regular intervals within the cluster. The sampling interval ($k$) was calculated by dividing the total number of HU's in the cluster by eight, which was the target number of HUs for enrollment. Field teams were assigned or selected a starting point in their cluster and approached every $k$th HU. If no adult was available at a selected HU, the field team revisited the HU at another time and/or day. HU's that did not meet eligibility criteria, where no adult was home after a second visit, or that declined to participate were replaced by approaching another $k$th house. Eligibility comprised the ability to give consent, age, ability to take/provide a nasal swab, and duration of residence at sampled address (Table 1). If eight HUs were not enrolled after completing a systematic sampling of every $k$th house, the field team selected a new starting point and approached every $k$th house from the new starting point. All individuals at a selected HU were invited to participate in the survey, provided they met the eligibility criteria. The study design was modeled after the World Health Organization immunization cluster survey design [23]. With this design, sampling eight HUs per cluster and sampling 20 clusters is anticipated to provide 10 percentage point precision on prevalence estimates of at least 75% (e.g., vaccination prevalence), and 5 percentage point

**Table 1. Eligibility criteria for housing unit and individual participation.**

| Housing Unit Exclusion Criteria | 1. Houses that are vacant, have signage that discourage soliciting or trespassing, or appear unsafe. |
| | 2. Group housing, such as dormitories and assisted living facilities. |
| | 3. Businesses, schools, places of worship, and other non-HU buildings. |
| | 4. No adult home after two visits. |
| | 5. Someone in the HU is in isolation or quarantine for SARS-CoV-2 |
| Participant Exclusion Criteria | 1. The selected HU is not their usual place of residence (they do not live there for at least 6 months in a calendar year). |
| | 2. Less than 2 years old |
| | 3. Incapable of providing informed consent |
| | 4. Unable to complete survey procedures, including self-collection of an anterior nares swab and completion of the questionnaires in English |

precision on prevalence estimates of 5% or less (e.g., infection prevalence) [23]. This design uses a minimum sample size of $n_{min} = DE \frac{z^2 p\ (1-p)}{d^2}$, where DE is the design effect (estimated to be 2), z is the z-score for the desired confidence level, p is the proportion, and d is the precision. The number of HUs per cluster (C) is $\frac{n_{min}}{C}$ [23].

This study was reviewed by the Cornell University Institutional Review Board (#2107010440) and the Cayuga Medical Center Institutional Review Board (#0821EP) and determined to not meet the definition of human participant research under 45 CFR 46.102(l)(2) because it was a limited public health surveillance activity supported by a public health authority. Participants provided written consent for sample and data collection; parents or guardians provided consent for minors to participate and completed the survey for them if necessary.

## Survey administration

A pilot survey was conducted in January 2022 with two field teams to test recruitment, enrollment, and data collection methods. Subsequently, three surveys were completed in Tompkins County: February 6th and 12th, 2022 in the City of Ithaca, New York; April 23rd and 24th, 2022 in the Town of Ithaca, New York; and October 22nd and 23rd, 2022 in the Town of Dryden, New York. Between seven and twelve field teams (groups of two or three staff members) were available for each survey. Each team surveyed two clusters. Field staff participated in virtual and in-person training and used a comprehensive field manual (https://doi.org/10.5281/zenodo.7750260).

## Sample collection

Participants self-collected an anterior nares swab by swirling a swab in each nostril for 10 seconds, observed by field staff. The swabs were transferred to tubes containing 1 mL of viral transport medium with the assistance of field staff, who wore appropriate personal protective equipment. Samples were kept on ice until they were transferred to a refrigerator at the end of the day and then transferred to the laboratory.

## Questionnaires

Survey participants completed study questionnaires (S1–S3) on a tablet via Qualtrics, or on paper if an internet connection was not available, in which case answers were transferred to the online questionnaires by field staff. Participants completed [1] a patient registration form, including their contact information, consent for SARS-CoV-2 PCR testing and the release of test results to the survey leaders, and demographic information (S4), and [2] a questionnaire to provide consent for serologic testing, and gather additional demographic information, vaccination and infection history, and data on behaviors and attitudes related to SARS-CoV-2 spread. Participants could elect to skip any question. The questionnaires were linked to the sample by identification number. Patient registration forms were maintained by the laboratory and not provided to the survey leaders.

## SARS-CoV-2 PCR testing

Testing for SARS-CoV-2 RNA was performed using a real-time reverse transcriptase RT-PCR assays targeting the nucleoprotein gene [24] at the Cornell COVID-19 Testing Laboratory [25]. The assay used for testing was validated for use in anterior nares swab samples. The sensitivity for pooled anterior nares specimens is 93% and the specificity for un-pooled specimens is 94%; specificity is expected to be the same for pooled and un-pooled specimens [24]. Participants received their test results from the laboratory.

## Serological testing

Serological testing and interpretation were conducted as previously described [26]. The multiplex assay has a nucleocapsid protein (NP) IgG sensitivity of 97.6% ($Se_{NP}$) and specificity of 100% ($Sp_{NP}$), and a receptor binding domain (RBD)

IgG sensitivity of 95.2% ($Se_{RBD}$) and specificity of 98.7% ($Sp_{RBD}$) [26]. The presence of RBD IgG alone was interpreted as evidence of vaccination without infection. The presence of NP IgG and RBD IgG was interpreted as evidence of infection, with or without vaccination. The serologic results were considered in parallel to determine overall seropositivity, from either infection or vaccination. The overall seropositivity sensitivity ($Se_{AB}$) and specificity ($Sp_{AB}$) were calculated accordingly: $Se_{AB} = 1 - (1 - Se_{NP}) * (1 - Se_{RBD})$; $Sp_{AB} = Sp_{NP} * SP_{RBD}$. Since the presence of NP IgG distinguishes from vaccination and natural infection, the $Se_{NP}$ and $Sp_{NP}$ were used in calculated the true prevalence of antibodies from infection.

### County-wide testing data and analysis

County-wide data on the number of PCR-positive tests and total number of PCR tests performed in the county on the dates of our survey, as well as cumulative total number of positive PCR cases up to and including the dates of the surveys, were extracted from the publicly available dashboard (S5–S7). Test results from county testing sites, healthcare facilities, and universities were included. County-level cumulative incidence (cumulative PCR-positive cases divided by total county population), point prevalence (number of PCR-positive cases on the date of the surveys divided by total county population), and test positivity (total number of PCR-positive tests divided by the total number of tests performed on the survey dates) were calculated.

### Survey data analysis

The data were imported into R (version 4.3.3) and analyzed using RStudio (version 2023.12.1). HUs were the sampling unit but the elementary unit is an individual. The proportion of positive tests per HU was calculated and the HU was weighted by $\frac{M_0 * P_h}{H * C}$, where $M_0$ is the number of total HU in the population, $P_h$ is the number of participants in the HU, $H$ is the number of HUs sampled in the population, and $C$ is the number of clusters sampled in the population. The sum of the HU weights represents the number of individuals in the population who would have participated in the study if every HU was sampled. The prevalence estimates for positive PCR tests and serology were adjusted for the survey design using the R package *survey*, with clustering at the sampled cluster and HU level and the HU weights used as sampling weights. The proportion of positive tests (apparent prevalence) and 95% confidence interval were calculated using a Rao-Scott likelihood method or a logistic regression and Wald-type interval if the likelihood method did not converge [27]. The true prevalence was calculated as $\frac{AP + Sp - 1}{Se + Sp - 1}$, where $AP$ is the apparent prevalence, $Sp$ is the test specificity, and $Se$ is the test sensitivity. The 95% confidence intervals were truncated at 0% and 100%.

PCR and serology results were matched to questionnaire responses using the sample tube identification numbers and patient registration numbers. Discrepancies in self-reported vaccination/infection and serology results were tabulated. Since the questionnaire responses could not easily be pooled at the HU level, only one adult per HU was randomly selected for analysis of the relationship between serology results, behaviors, and attitudes. Partially complete questionnaires were included. The HU questionnaire response was weighted using the same approach as the test results above and descriptive statistics were adjusted for the survey design. The association between seropositivity and self-reported behaviors and attitudes was assessed with Kruskal-Wallis tests for continuous variables, Wald tests for categorical variables, and Wilcoxon rank-sum tests for ordinal variables.

## Results

### Participants

Overall, 221 HUs participated in the surveillance surveys (Fig 1). Field teams approached 928 HUs and the door was answered at 583 HUs (approximately 60% of the HUs approached in all surveys). A small number of HUs were excluded (Fig 1). HU participation proportions were 43.5% (February), 36.4% (April), and 43.3% (October).

The average, minimum, and maximum number of HUs enrolled per cluster for each survey is shown in Fig 1. In general, one person per HU participated in the surveys, although at some HUs multiple people elected to participate. In

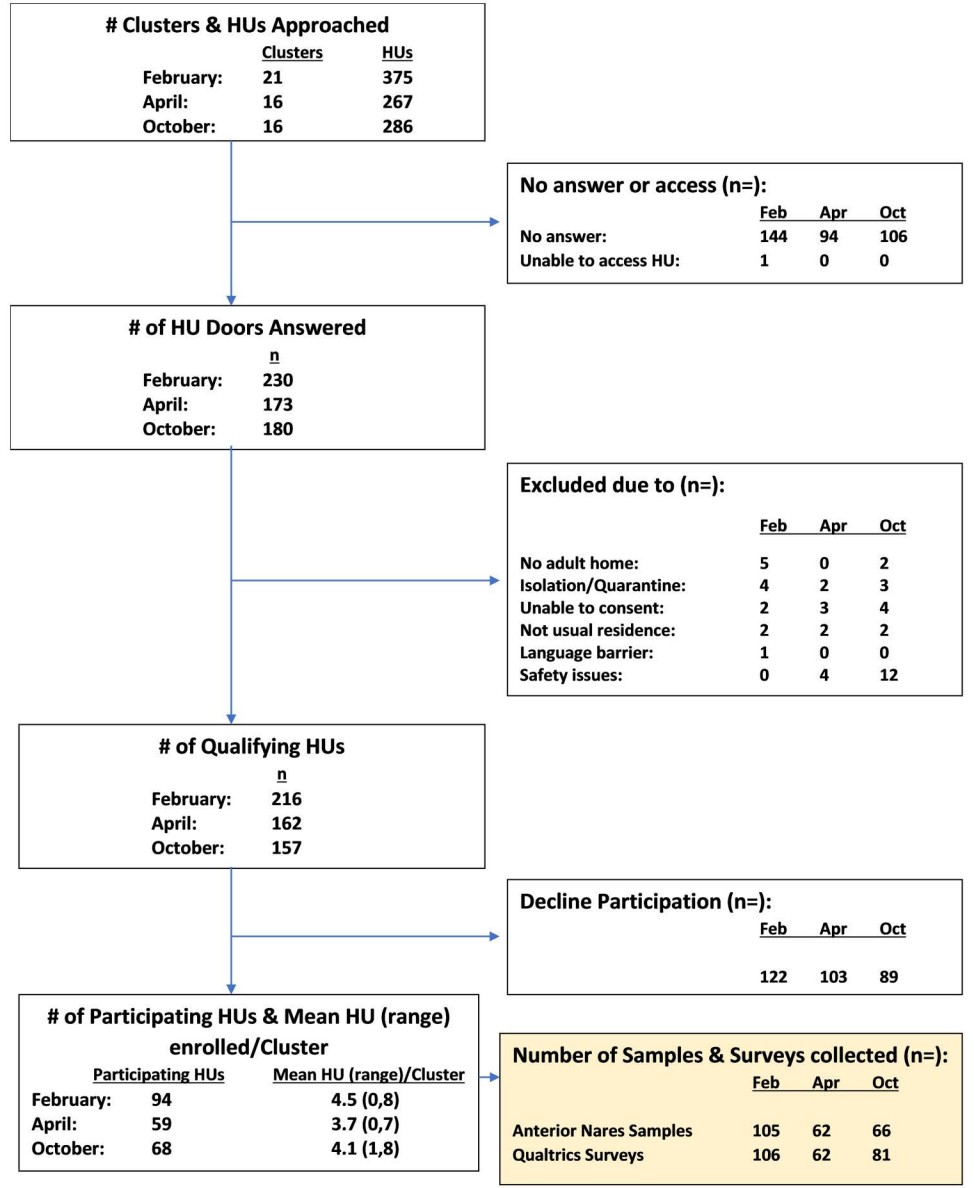

**Fig 1. Housing Units (HUs) approached, excluded, and participating by survey month.**

some cases, participants chose to complete the questionnaire but did not provide an anterior nares sample or vice versa. In February, 105 participants provided an anterior nares sample and 101 could be linked to a questionnaire response. In April, 62 samples were collected and 57 were linked to a questionnaire response; and in October 66 samples were collected and 57 were associated with a questionnaire. All samples were used in infection and immunity prevalence estimates, but only samples that could be linked to a questionnaire response were used for the rest of the analysis. After correcting for sampling methodology and selecting one adult per HU, the demographics of survey participants were similar to the overall municipality demographics as measured in the 2020 U.S. Census [28], considering the small sample size (Table 2). Our sample population included slightly more females, older individuals, and white individuals than all three communities (City of Ithaca, Town of Ithaca, Town of Dryden).

**Table 2. Survey participant demographics compared to US Census demographics. Survey demographics were adjusted for sampling methodology and included only one adult per HU.**

| Demographic | | City of Ithaca | | Town of Ithaca | | Town of Dryden | |
|---|---|---|---|---|---|---|---|
| | | Survey | Census | Survey | Census | Survey | Census |
| Race | White alone | 76% | 68.0% | 88% | 74.4% | 91% | 89.9% |
| | African American alone | 1.7% | 5.7% | 1.3% | 5.3% | 0% | 1.6% |
| | Asian | 16% | 16.1% | 7.1% | 11.8% | 3.2% | 3.5% |
| | American Indian or Alaska Native alone | 0% | 0.1% | 0% | 0.1% | 2.3% | 0.1% |
| | Native Hawaiian or Other Pacific Islander alone | 0% | 0.1% | 0% | 0.1% | 0% | 0% |
| | Two or more races | 5.6% | 6.8% | 0% | 5.1% | 0% | 4% |
| Ethnicity | Hispanic or Latino | 3% | 8.4% | 3.7% | 6.0% | 4.8% | 3.4% |
| Gender | Female | 53% | 49.2% | 61% | 52.8% | 56% | 49.4% |
| Age | 65 years and over | 17% | 7.8% | 21% | 19.9% | 33% | 18.4% |

## Infection and antibody prevalence

In the City of Ithaca in February 2022, only two out of 105 samples were positive on SARS-CoV-2 PCR testing (test positivity of 1.9%). Accounting for the test sensitivity and specificity, the estimated true prevalence of SARS-CoV-2 infection in the City of Ithaca in February was less than 2% of individuals were infected (95% confidence interval: 0%, 2.0%). The true prevalence of prior infection, as measured by the presence of antibodies against NP, was 49% of individuals (95% confidence interval: 38%, 60%). Overall, 92% (95% confidence interval: 86%, 99%) of the City of Ithaca individuals had detectable antibodies from either infection or vaccination at the time of the survey.

Similarly, only two out of 62 RT-PCR tests were positive in April 2022 in the Town of Ithaca (test positivity of 3.2%), resulting in an estimated true prevalence of less than 5% (95% confidence interval: 0%, 4.4%). The estimated true prevalence of prior infection was 45% (95% confidence interval: 33%, 57%). Overall, 90% (95% confidence interval: 77%, 100%) of the Town of Ithaca individuals had detectable antibodies from either infection or vaccination at the time of the survey.

In October 2022, there were no positive PCR tests out of 66 samples in our survey of the Town of Dryden. Similar to the other municipalities and time points, 45% of individuals (95% confidence interval: 30%, 59%) had antibodies from prior infection. Prevalence of antibodies from infection and/or vaccination were lower compared to other municipalities: 81% (95% confidence interval: 72%, 90%).

Test positivity rates of our survey, not adjusted for survey design, were 1.9%, 3.2%, and 0% in February, April, and October, respectively, less than the overall county-wide seven-day average test positivity rates of 2.5%, 6.3%, and 9.0%, respectively, for the survey time periods. The expected point prevalence of PCR positive tests was calculated from the county-level reported new cases per day, assuming that infected individuals were PCR positive for 14 days [29], and the county population [28]. The overall county expected PCR-positive point prevalences (1.0% in February, 0.7% in April, 0.3% in October) based on reported PCR-positive test results, and point prevalences based on reported positive PCR and at-home antigen tests (1.3% in February, 1% in April, 0.4% in October), were generally within the survey estimates. Estimated county level cumulative infection prevalences, based on the reported number of unique individuals with positive PCR tests, were 16.6% in February, 19.1% in April, and 23.8% in October, and were 18.2% in February, 21.3% in April, and 27.7% in October based on positive PCR and at-home antigen tests. These cumulative prevalences are substantially lower than the prevalence of individuals with antibodies from infection in the surveys.

## Self-reported vs. detected infection and vaccination

Serologic testing for antibodies confirmed that most survey participants had antibodies from vaccination and/or prior infection. The population-level prevalence of primary vaccination was high in all three communities: 97% in the City of Ithaca in

February, 100% in the Town of Ithaca in April, and 93% in the Town of Dryden in October. Overall, 7% (n = 15/203) of participants who reported being fully vaccinated did not have detectable antibodies (Table 3). The October 2022 survey had a higher proportion of vaccinated participants without antibodies (15%, 8/54, Table 3), and had the lowest population-level prevalence of booster vaccination among already-vaccinated individuals (80% October, 93% April, 95% February).

Substantially more participants had antibodies from a prior infection than the number of participants who self-reported an infection. Only 31% (February), 33% (April), and 50% (October) of people who had antibodies from infection reported having a previous positive test for COVID-19. The proportion of people without antibodies from infection who self-reported a prior positive test increased over time (11% February, 22% April, 58% October).

## Behaviors and attitudes

Participants were asked about behaviors and attitudes that could be associated with risk of SARS-CoV-2 infection, including travel, social distancing, gathering with other people, masking, smoking and vaping, hand-sanitizing and other related behaviors, going to doctor's appointments, using telehealth, and working environment. In univariate comparisons, few factors were associated with seropositivity for a past infection at an alpha = 0.05 (February: taking the metro, use of telehealth services; April: smoking, frequency of cleaning behaviors, use of telehealth services, perception of the importance of masking and travel; October: none; S9–S20). Due to the small sample size, multivariable models of seropositivity that included all potential risk factors and confounders could not be built.

## Discussion

Accurate and timely surveillance of SARS-CoV-2 prevalence and immunity is critical to local and national COVID-19 pandemic responses. Representative surveillance surveys reveal more accurate estimates of COVID-19 infection than other measures based on reported test results. We set out to provide local health department officials with prevalence estimates calculated from a representative sample to better inform their decision-making efforts in response to the COVID-19 pandemic, and to identify characteristics associated with COVID-19 infections among high-risk groups. While our sample did not provide robust details to elucidate characteristics associated with higher risk, we did find that the survey estimates of prior infection based on seropositivity were two to three times higher than the expected prevalence from cumulative unique positive tests at the county level. Higher rates of infection from representative samples compared to reported tests have been found in other studies as well [3,4,9–15], primarily because representative samples capture asymptomatic and symptomatic infections whereas tests are conducted primarily among symptomatic individuals [9–11,30,31]. One study reported that among a random sample in Iran, where cluster sampling was used to recruit a representative sample, 57% of SARS-CoV-2- infected individuals were asymptomatic [9]. In this study, fewer than 50% of people with antibodies from past infections reported a previous positive test, suggesting a similar proportion of asymptomatic infections (Table 4). Alternatively,

**Table 3. Discrepancies between self-reported vaccination status and seropositivity. Number of participants is reported in each cell. Partially vaccinated indicates the participant had one vaccine of a two-vaccine series.**

| Self-reported Vaccination | | Antibody Interpretation | | |
|---|---|---|---|---|
| | | Prior Infection +/- Vaccination | Vaccination without Infection | Negative |
| February, City of Ithaca | No | 0 | 0 | 2 |
| | Partially | 0 | 0 | 2 |
| | Yes | 45 | 46 | 4 |
| April, Town of Ithaca | No | 0 | 0 | 2 |
| | Yes | 24 | 27 | 3 |
| October, Town of Dryden | No | 0 | 0 | 3 |
| | Yes | 26 | 20 | 8 |

**Table 4. Discrepancies between self-reported positive tests and seropositivity.**

| Self-reported Previous Positive Test | | Antibodies from Infection | |
| --- | --- | --- | --- |
| | | Negative | Positive |
| **February, City of Ithaca** | No | 49 | 31* |
| | Yes | 6 | 14 |
| **April, Town of Ithaca** | No | 25 | 16† |
| | Yes | 7 | 8 |
| **October, Town of Dryden** | No | 13 | 13‡ |
| | Yes | 18 | 13 |

*Two participants reported thinking they had been infected but not having a positive test and six reported being unsure if they had been infected

†Five participants reported being unsure if they had been infected but not had a positive test

‡Four participants reported being unsure if they had been infected but not had a positive test

the discrepancy between previous positive test results and seroprevalence could be due to insufficient access to testing. Testing programs were available in Tompkins County through the local hospital, health department, and universities [22,32], however availability of testing does not preclude other barriers to access, such as those based on beliefs or socioeconomic barriers (e.g., ability to get time off work to get a test) [5]. Finally, the lower test positivity in the surveys compared to county-wide data would be expected due to differences in the testing programs (i.e., entirely random testing in the survey vs county-wide data including tests from individuals seeking care and more likely to be infected).

Reports vary for prevalence of SARS-CoV-2 infection and immunity of representative populations around the United States during the pandemic; two studies reported prevalences over the same time frame as these surveys of Tompkins County [3,16]. In February 2022, approximately 58% of a convenience sample of laboratory blood specimens had anti-NP antibodies, indicating past infection [3], similar to the prevalence estimate for the City of Ithaca (49%, 95% confidence interval: 38%, 60%). Approximately half the U.S. population had been exposed to SARS-CoV-2 by early 2022. A study of blood donors estimated that the U.S. population prevalence of any antibodies, from vaccination or infection, was over 90% by January – March 2022 [16], similar to the estimates from these surveys. During the surveys, the predominant variants in the U.S. Health and Human Services Region 2, which includes New York State, were B.1.1.529 (February 2022), BA.2 (April 2022), and BA.5 (October 2022) [33].

Notably, the survey in October 2022 had the lowest prevalence of overall seropositivity, but similar levels of antibodies from past infections to the other time points, suggesting a lower prevalence of vaccination or greater waning of vaccination-derived antibodies in the October 2022 participants. Antibody titers can be affected by waning of neutralizing antibody levels post-infection [30,31] and post-vaccination [34,35]. The prevalence of primary vaccination was slightly lower in the Town of Dryden (October 2022) than the other survey locations but was still over 90%. However, the booster vaccination prevalence was lowest in the October survey. In addition, the proportion of people who had reported a previous positive test but did not have detectable NP antibodies increased over time (30% February, 47% April, 58% October), suggesting waning antibodies from infection [30,31]. Therefore, decreasing antibody titers from both infection and vaccination likely contributed to the overall lower seroprevalence in October 2022.

These surveys are limited by a small sample size, including failing to recruit the targeted eight HUs per cluster, and by collecting data at only one time point per municipality. The cluster preview process likely resulted in some selection bias, since neighborhoods with a large proportion of individuals in group homes (e.g., dormitories or assisted living facilities) and those with a lot of HUs with "no solicitation" signs (mainly apartment buildings and manufactured home communities) were excluded. Group homes and the communities with "no solicitation" signs were generally high-density communities, which could lead to an under-estimation of SARS-CoV-2 prevalence since SARS-CoV-2 would likely spread more in high-density compared to low-density communities. In addition, there is likely non-response bias since individuals who were less concerned with SARS-CoV-2 may have been less likely to participate, and also less likely to be vaccinated or practice preventive measures. On the other hand, those who are more concerned with SARS-CoV-2 (and more likely to

be vaccinated and practice preventive measures) may have been more interested in participating to receive a SARS-CoV-2 test and/or contribute to research. These sources of response bias may lead to underestimation of SARS-CoV-2 prevalence. In these surveys, the effect of time and location cannot be separated. Due to the small sample size, the associations between potential risk factors, including behaviors, attitudes, and participant characteristics, and previous or current infection were not explored. However, the sample was generally representative of the municipalities surveyed, and prevalence estimates were consistent with other nationwide surveys conducted at similar times. Participants' recall of positive test results and vaccination status may introduce recall and misclassification bias [10,16].

## Conclusions

Accurate estimates of prevalence of SARS-CoV-2 infection and immunity were determined and provided in a timely fashion to local public health officials in Tompkins County, NY to help inform their pandemic decision-making. The prevalence of SARS-CoV-2 infection as assessed by NP antibodies was two to three times higher than an estimated prevalence based on reported positive tests, highlighting the substantial burden of asymptomatic or mild undiagnosed infections and spread. Large samples are required to assess the effect of SARS-CoV-2 preventive behaviors and attitudes on infection spread. Despite being costly and time-intensive, representative surveillance surveys provide accurate measures of population-based infection and immunity rates.

## Supporting information

**S1 File. Qualtrics surveys administered in February 2022.**
(PDF)

**S2 File. Qualtrics surveys administered in April 2022.**
(PDF)

**S3 File. Qualtrics surveys administered in October 2022.**
(PDF)

**S4 File. Laboratory patient registration form with demographic questions.**
(DOCX)

**S5 File. County case data extracted from the Tompkins County Health Department public COVID-19 dashboards on cumulative cases.**
(CSV)

**S6 File. County case data extracted from the Tompkins County Health Department public COVID-19 dashboards on new cases.**
(CSV)

**S7 File. County case data extracted from the Tompkins County Health Department public COVID-19 dashboards on positive tests.**
(CSV)

**S8 File. STROBE checklist.**
(DOCX)

**S9 Table. Apr_attitudes_x_InfectAB_weighted.** Table of the univariate comparisons between antibody presence and attitude risk factors for infection in April 2022.
(HTML)

**S10 Table. Apr_behavior_x_InfectAB_weighted.** Table of the univariate comparisons between antibody presence and behavior risk factors for infection in April 2022.
(HTML)

**S11 Table. Apr_demographics_x_InfectAB_weighted.** Table of the univariate comparisons between antibody presence and demographic risk factors for infection in April 2022.
(HTML)

**S12 Table. Apr_work_x_InfectAB_weighted.** Table of the univariate comparisons between antibody presence and work risk factors for infection in April 2022.
(HTML)

**S13 Table. Feb_attitudes_x_InfectAB_weighted.** Table of the univariate comparisons between antibody presence and attitude risk factors for infection in February 2022.
(HTML)

**S14 Table. Feb_behavior_x_InfectAB_weighted.** Table of the univariate comparisons between antibody presence and behavior risk factors for infection in February 2022.
(HTML)

**S15 Table. Feb_demographics_x_InfectAB_weighted.** Table of the univariate comparisons between antibody presence and demographic risk factors for infection in February 2022.
(HTML)

**S16 Table. Feb_work_x_InfectAB_weighted.** Table of the univariate comparisons between antibody presence and work risk factors for infection in February 2022.
(HTML)

**S17 Table. Oct_attitudes_x_InfectAB_weighted.** Table of the univariate comparisons between antibody presence and attitude risk factors for infection in October 2022.
(HTML)

**S18 Table. Oct_behavior_x_InfectAB_weighted.** Table of the univariate comparisons between antibody presence and behavior risk factors for infection in October 2022.
(HTML)

**S19 Table. Oct_demographics_x_InfectAB_weighted.** Table of the univariate comparisons between antibody presence and demographic risk factors for infection in October 2022.
(HTML)

**S20 Table. Oct_work_x_InfectAB_weighted.** Table of the univariate comparisons between antibody presence and work risk factors for infection in October 2022.
(HTML)

## Acknowledgments

This survey would not have been possible without the help of the field staff and community partners, including Cayuga Health Partners, the Cornell COVID-19 Testing Laboratory, Tompkins County Health Department, and local police departments. The authors would also like to thank all the survey participants for giving so generously of their time.

## Author contributions

**Conceptualization:** Casey L. Cazer, Bettina Wagner, Diego G. Diel, Katherine R. McLaughlin, Jeffrey W. Bethel, Kevin J. Cummings, Genevive R. Meredith, Benjamin D. Dalziel.

**Data curation:** Casey L. Cazer, Jeanne W. Lawless, Parshad Mehta, Bettina Wagner, Diego G. Diel.

**Formal analysis:** Casey L. Cazer, Jeanne W. Lawless, Parshad Mehta.

**Funding acquisition:** Casey L. Cazer, Katherine R. McLaughlin, Jeffry W. Bethel, Benjamin D. Dalziel.

**Investigation:** Casey L. Cazer, Jeanne W. Lawless, Parshad Mehta, Bettina Wagner, Diego G. Diel, Katherine R. McLaughlin, Jeffrey W. Bethel, Elizabeth F. Plocharczyk, Genevive R. Meredith, Benjamin D. Dalziel.

**Methodology:** Casey L. Cazer, Bettina Wagner, Diego G. Diel, Katherine R. McLaughlin, Jeffrey W. Bethel, Kevin J. Cummings, Genevive R. Meredith, Samantha Hillson, Robert Lawlis, Lara Parrilla, Benjamin D. Dalziel.

**Project administration:** Casey L. Cazer, Jeanne W. Lawless, Katherine R. McLaughlin, Jeffrey W. Bethel, Benjamin D. Dalziel.

**Resources:** Casey L. Cazer, Bettina Wagner, Diego G. Diel, Katherine R. McLaughlin, Jeffrey W. Bethel, Elizabeth F. Plocharczyk, Robert Lawlis, Benjamin D. Dalziel.

**Supervision:** Casey L. Cazer, Katherine R. McLaughlin, Jeffrey W. Bethel, Benjamin D. Dalziel.

**Visualization:** Jeanne W. Lawless.

**Writing – original draft:** Casey L. Cazer, Jeanne W. Lawless, Parshad Mehta.

**Writing – review & editing:** Casey L. Cazer, Jeanne W. Lawless, Parshad Mehta, Bettina Wagner, Diego G. Diel, Katherine R. McLaughlin, Jeffrey W. Bethel, Elizabeth F. Plocharczyk, Kevin J. Cummings, Genevive R. Meredith, Samantha Hillson, Robert Lawlis, Lara Parrilla, Benjamin D. Dalziel.

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
