## [Decision Letter · Decision Letter 0]

13 Nov 2024

Dear Dr. Cazer,

Thank you for submitting your manuscript to PLOS ONE. After careful consideration, we feel that it has merit but does not fully meet PLOS ONE’s publication criteria as it currently stands. Therefore, we invite you to submit a revised version of the manuscript that addresses the points raised during the review process.

We look forward to receiving your revised manuscript.

Kind regards,

Harapan Harapan, MD, PhD

Academic Editor

PLOS ONE

Journal Requirements: When submitting your revision, we need you to address these additional requirements. 1. Please ensure that your manuscript meets PLOS ONE's style requirements, including those for file naming. The PLOS ONE style templates can be found at https://journals.plos.org/plosone/s/file?id=wjVg/PLOSOne_formatting_sample_main_body.pdf and https://journals.plos.org/plosone/s/file?id=ba62/PLOSOne_formatting_sample_title_authors_affiliations.pdf 2. Thank you for stating the following financial disclosure: "This work was supported by a grant (#2021-72608) from the David and Lucile Packard Foundation."  Please state what role the funders took in the study.  If the funders had no role, please state: ""The funders had no role in study design, data collection and analysis, decision to publish, or preparation of the manuscript."" If this statement is not correct you must amend it as needed. Please include this amended Role of Funder statement in your cover letter; we will change the online submission form on your behalf. 3. We note that you have indicated that there are restrictions to data sharing for this study. For studies involving human research participant data or other sensitive data, we encourage authors to share de-identified or anonymized data. However, when data cannot be publicly shared for ethical reasons, we allow authors to make their data sets available upon request. For information on unacceptable data access restrictions, please see http://journals.plos.org/plosone/s/data-availability#loc-unacceptable-data-access-restrictions.  Before we proceed with your manuscript, please address the following prompts: a) If there are ethical or legal restrictions on sharing a de-identified data set, please explain them in detail (e.g., data contain potentially identifying or sensitive patient information, data are owned by a third-party organization, etc.) and who has imposed them (e.g., a Research Ethics Committee or Institutional Review Board, etc.). Please also provide contact information for a data access committee, ethics committee, or other institutional body to which data requests may be sent. b) If there are no restrictions, please upload the minimal anonymized data set necessary to replicate your study findings to a stable, public repository and provide us with the relevant URLs, DOIs, or accession numbers. Please see http://www.bmj.com/content/340/bmj.c181.long for guidelines on how to de-identify and prepare clinical data for publication. For a list of recommended repositories, please see https://journals.plos.org/plosone/s/recommended-repositories. You also have the option of uploading the data as Supporting Information files, but we would recommend depositing data directly to a data repository if possible. Please update your Data Availability statement in the submission form accordingly. 4. Please review your reference list to ensure that it is complete and correct. If you have cited papers that have been retracted, please include the rationale for doing so in the manuscript text, or remove these references and replace them with relevant current references. Any changes to the reference list should be mentioned in the rebuttal letter that accompanies your revised manuscript. If you need to cite a retracted article, indicate the article’s retracted status in the References list and also include a citation and full reference for the retraction notice.

Reviewers' comments:

Reviewer's Responses to Questions

**Comments to the Author**

1. Is the manuscript technically sound, and do the data support the conclusions?

Reviewer #1: Yes

Reviewer #2: Yes

2. Has the statistical analysis been performed appropriately and rigorously?

Reviewer #1: Yes

Reviewer #2: Yes

3. Have the authors made all data underlying the findings in their manuscript fully available?

Reviewer #1: Yes

Reviewer #2: Yes

4. Is the manuscript presented in an intelligible fashion and written in standard English?

Reviewer #1: Yes

Reviewer #2: Yes

Reviewer #1: A review of the manuscript entitled “Prevalence of SARS-CoV-2 infection and immunity in a New York county in 2022 reveals frequent asymptomatic or undiagnosed infections”

1. (page 5, lines 48-49): This sentence “Disease surveillance is a cornerstone of public health practice, particularly to understand and control the spread of infectious pathogens” will be more significant if it’s also supported by another relevant study. Please check “Global prevalence of persistent neuromuscular symptoms and the possible pathomechanisms in COVID-19 recovered individuals: A systematic review and meta-analysis”

2. (page 11, lines 147-150): The sentence also shares the same idea with a study entitled “Comparison of RT-PCR cycle threshold values between individual and pooled SARS-CoV-2 infected nasopharyngeal swab specimens” Kindly include this reference to make the sentence stronger.

3. (page 14, lines 216-218): The authors stated that “Our sample population included slightly more females, older individuals, and white individuals than all three communities.” My question is what are those three communities? Please briefly mention them in this sentence.

4. (page 16, lines 223-224): The sentence has a few grammatical issues that need correction. Please consider omitting the “were positive” part to make the sentence clear. Thus, the final sentence looks like this “In the City of Ithaca in February 2022, only two out of 105 samples on SARS-CoV-2 PCR testing was positive (test positivity of 1.9%).”

5. Overall, this manuscript has shown a sophisticated review of the estimation of SARS-CoV-2 prevalence in Tompkins County, using lots of relevant references in related areas. The use of good English has also been applied throughout the manuscript. However, I have some minor comments, as already mentioned above, to increase the clarity of this review manuscript. Thank you.

Reviewer #2: Thank you for the opportunity to review this well-written manuscript. The study by Cazer et al. estimated prevalence of COVID-19 infection from population-representative household surveys in Tompkins County. The authors note that the use of indicators such as percent positivity from case based surveillance are biased and are likely underestimating the true prevalence of infection in the community. The study sampled households in three municipalities at one time point for each.

In summary, the study found that the prevalence of active infection was less than 5%, about half of the population had a prior infection, and a large proportion of survey participants had hybrid immunity due to either prior infection of vaccination. Authors note substantial proportion of likely asymptomatic infections and waning immunity post-vaccination and post-infection over time especially in the survey administered in Oct 2022. Stated strengths include probability-based sampling, and weaknesses include small sample size that precluded analyses looking at behaviors and previous and current infections.

Main comments:

1.While probability-based sampling in population representative surveys can reduce bias, population representative surveys are not completely impervious to selection bias which can occur at different stages of sampling and inclusion (i.e., who is invited and who responds/participates in survey). Limitations on the possible sources of bias should be transparent and included in the discussion. For example, it is important to note that the HU sampling scheme excludes group homes, dormitories and assisted living facilities, which limits the external validity of the findings especially when compared to cases reported through surveillance (which presumably captures cases from those places). Further, non-response bias may have occurred among those who agreed to consent to the survey. If those who responded are more or less likely to have been previously infected or previously vaccinated, then it could affect your prevalence estimates. The directionality of the bias could either over- or under-estimate the prevalence but noting this potential source of bias is important in the discussion.

2.Was there information available on those who declined participation? If so, were they compared with those who chose to participate? How were they different?

3.Did participants get access to their test results? If so, motivations to get tested in order to obtain their test results may bias the estimates (e.g., those who were symptomatic are more likely to get tested to confirm test result, and therefore may be over-representative). Would suggest incorporating additional assessment of potential sources of selection bias in the discussion to address the points above.

Other:

4.Were sample size calculations done for this study? Would suggest including the decision around the number of HU per cluster sampled.

5.The paper would benefit from a brief situation analysis of circulating variants during the survey time periods.

6.Introduction, line 53; would include the use of rapid at-home diagnostic testing as it would have contributed to individuals who tested at home and did not seek confirmatory testing with a provider and therefore led to an under count of cases.

7.Did county-level metrics only include PCR tests and test results? Some jurisdictions also include probable cases based on rapid antigen tests. If those were available, would consider presenting those numbers as well by comparing the survey-based estimates with PCR (confirmed) and rapid antigen test (probable).

**Do you want your identity to be public for this peer review?** For information about this choice, including consent withdrawal, please see our Privacy Policy

Reviewer #1: No

Reviewer #2: No

---

## [Author Response · Author response to Decision Letter 0]

25 Nov 2024

Reviewer #1: A review of the manuscript entitled “Prevalence of SARS-CoV-2 infection and immunity in a New York county in 2022 reveals frequent asymptomatic or undiagnosed infections”

1. (page 5, lines 48-49): This sentence “Disease surveillance is a cornerstone of public health practice, particularly to understand and control the spread of infectious pathogens” will be more significant if it’s also supported by another relevant study. Please check “Global prevalence of persistent neuromuscular symptoms and the possible pathomechanisms in COVID-19 recovered individuals: A systematic review and meta-analysis”

Authors: The sentence is intentionally broad and not specific to COVID-19, hence we did not cite a COVID-19 paper but rather an overview of public health surveillance. We decline to include this reference in this section.

2. (page 11, lines 147-150): The sentence also shares the same idea with a study entitled “Comparison of RT-PCR cycle threshold values between individual and pooled SARS-CoV-2 infected nasopharyngeal swab specimens” Kindly include this reference to make the sentence stronger.

Authors: The reference we included (Laverack M, Tallmadge RL, Venugopalan R, Cronk B, Zhang X, Rauh R, et al. Clinical evaluation of a multiplex real-time RT-PCR assay for detection of SARS-CoV-2 in individual and pooled upper respiratory tract samples. Arch Virol. 2021;166(9):2551-61) was a description of the actual test used in our study. We do not want to confuse readers and cite an assay that we did not use. Therefore, we decline to include this reference here.

3. (page 14, lines 216-218): The authors stated that “Our sample population included slightly more females, older individuals, and white individuals than all three communities.” My question is what are those three communities? Please briefly mention them in this sentence.

Authors: The specific communities were added (line 223).

4. (page 16, lines 223-224): The sentence has a few grammatical issues that need correction. Please consider omitting the “were positive” part to make the sentence clear. Thus, the final sentence looks like this “In the City of Ithaca in February 2022, only two out of 105 samples on SARS-CoV-2 PCR testing was positive (test positivity of 1.9%).”

Authors: This sentence was corrected (line 229).

5. Overall, this manuscript has shown a sophisticated review of the estimation of SARS-CoV-2 prevalence in Tompkins County, using lots of relevant references in related areas. The use of good English has also been applied throughout the manuscript. However, I have some minor comments, as already mentioned above, to increase the clarity of this review manuscript. Thank you.

Reviewer #2: Thank you for the opportunity to review this well-written manuscript. The study by Cazer et al. estimated prevalence of COVID-19 infection from population-representative household surveys in Tompkins County. The authors note that the use of indicators such as percent positivity from case based surveillance are biased and are likely underestimating the true prevalence of infection in the community. The study sampled households in three municipalities at one time point for each.

In summary, the study found that the prevalence of active infection was less than 5%, about half of the population had a prior infection, and a large proportion of survey participants had hybrid immunity due to either prior infection of vaccination. Authors note substantial proportion of likely asymptomatic infections and waning immunity post-vaccination and post-infection over time especially in the survey administered in Oct 2022. Stated strengths include probability-based sampling, and weaknesses include small sample size that precluded analyses looking at behaviors and previous and current infections.

Main comments:

1. While probability-based sampling in population representative surveys can reduce bias, population representative surveys are not completely impervious to selection bias which can occur at different stages of sampling and inclusion (i.e., who is invited and who responds/participates in survey). Limitations on the possible sources of bias should be transparent and included in the discussion. For example, it is important to note that the HU sampling scheme excludes group homes, dormitories and assisted living facilities, which limits the external validity of the findings especially when compared to cases reported through surveillance (which presumably captures cases from those places). Further, non-response bias may have occurred among those who agreed to consent to the survey. If those who responded are more or less likely to have been previously infected or previously vaccinated, then it could affect your prevalence estimates. The directionality of the bias could either over- or under-estimate the prevalence but noting this potential source of bias is important in the discussion.

Authors: A discussion of selection bias and non-response bias has been added (lines 346-359).

2.Was there information available on those who declined participation? If so, were they compared with those who chose to participate? How were they different?

Authors: There is no information available on individuals who declined to participate, had “no solicitation signs”, or were not home at the time of the survey. A discussion of selection bias and non-response bias has been added (lines 346-359).

3.Did participants get access to their test results? If so, motivations to get tested in order to obtain their test results may bias the estimates (e.g., those who were symptomatic are more likely to get tested to confirm test result, and therefore may be over-representative). Would suggest incorporating additional assessment of potential sources of selection bias in the discussion to address the points above.

Authors: A discussion of selection bias and non-response bias has been added (lines 346-359). Participants did receive their test results; this was added to line 155.

Other:

4.Were sample size calculations done for this study? Would suggest including the decision around the number of HU per cluster sampled.

Authors: Details on the power of the study design were added to lines 109-111.

5.The paper would benefit from a brief situation analysis of circulating variants during the survey time periods.

Authors: This information was added (lines 330-332).

6.Introduction, line 53; would include the use of rapid at-home diagnostic testing as it would have contributed to individuals who tested at home and did not seek confirmatory testing with a provider and therefore led to an under count of cases.

Authors: This was added to line 54.

7.Did county-level metrics only include PCR tests and test results? Some jurisdictions also include probable cases based on rapid antigen tests. If those were available, would consider presenting those numbers as well by comparing the survey-based estimates with PCR (confirmed) and rapid antigen test (probable).

Authors: The reported county-level metrics only included PCR tests. We added the PCR and rapid antigen tests to lines 254-259. We cannot use the rapid antigen tests in the positive test prevalence since the denominator of rapid tests performed is unknown.

---

## [Editor Report · Decision Letter 1]

4 Dec 2024

Dear Dr. Cazer,

Thank you for submitting your manuscript to PLOS ONE. After careful consideration, we feel that it has merit but does not fully meet PLOS ONE’s publication criteria as it currently stands. Therefore, we invite you to submit a revised version of the manuscript that addresses the points raised during the review process.

The revised version of this manuscript should prioritize the following points:

Please include the limitations on the sources' bias in the discussion.Provide information on the sample size calculation.The manuscript should have the writing errors corrected.

We look forward to receiving your revised manuscript.

Kind regards,

Harapan Harapan, MD, PhD

Academic Editor

PLOS ONE
---

## [Author Response · Author response to Decision Letter 1]

3 Jan 2025

1. Please include the limitations on the sources' bias in the discussion.

Authors: A discussion of selection bias and non-response bias is in the discussion (lines 346-359). We have included the directionality of the bias (potential under-estimation of prevalence, lines 351-352 and lines 358-359). If there are additional biases that you would like to see discussed, please provide specifics.

2. Provide information on the sample size calculation.

Authors: Details on the sample size calculation and power of the study design are in lines 109-113. The sample size was calculated based on prevalence precision. The equation was added (lines 113-116).

3. The manuscript should have the writing errors corrected.

Authors: The manuscript was thoroughly reviewed for typos and grammatical errors. Please advise if specific errors are identified.

---

## [Decision Letter · Decision Letter 2]

6 Mar 2025

Dear Dr. Cazer,

Thank you for submitting your manuscript to PLOS ONE. After careful consideration, we feel that it has merit but does not fully meet PLOS ONE’s publication criteria as it currently stands. Therefore, we invite you to submit a revised version of the manuscript that addresses the points raised during the review process.

We look forward to receiving your revised manuscript.

Kind regards,

Harapan Harapan, MD, PhD

Academic Editor

PLOS ONE

Journal Requirements:

Reviewers' comments:

Reviewer's Responses to Questions

**Comments to the Author**

Reviewer #1: All comments have been addressed

Reviewer #3: All comments have been addressed

2. Is the manuscript technically sound, and do the data support the conclusions?

Reviewer #1: Yes

Reviewer #3: Yes

3. Has the statistical analysis been performed appropriately and rigorously?

Reviewer #1: Yes

Reviewer #3: Yes

4. Have the authors made all data underlying the findings in their manuscript fully available?

Reviewer #1: Yes

Reviewer #3: No

5. Is the manuscript presented in an intelligible fashion and written in standard English?

Reviewer #1: Yes

Reviewer #3: Yes

Reviewer #1: (No Response)

Reviewer #3: 1. On page 6 lines 72-74, I suggest that the authors add an additional group of people other than those in university settings, for example, the global survivors of COVID-19. Thus, the sentence becomes like this: “Finally, surveillance for behavioral factors that mitigate or promote COVID-19 infection risk is important in developing public health recommendations targeting groups at high risk (17-20), including in university settings (21) and the global survivors of COVID-19 (36).” The 36th reference can be cited from a study by Fahriani et al. entitled “Persistence of long COVID symptoms in COVID-19 survivors worldwide and its potential pathogenesis -A systematic review and meta-analysis”.

2. This sentence on page 6 lines 74-76 will be deeper if authors could provide an example of the settings they mention before, such as healthcare professional settings. Thus the sentence will look like this “Integrating surveillance for active infection with surveillance for preventive behaviors can be successful in managing COVID-19 infection in some settings (22), such as the healthcare professional settings (37).” This 37th reference can be found in a study by Hamdan et al. entitled “Coping strategies used by healthcare professionals during COVID-19 pandemic in Dubai: A descriptive cross-sectional study”.

3. On page 9 table 1, the authors stated that one of the conditions that a participant was excluded from this study was being less than 2 years old. Does it mean that authors include minors starting from 3 years old and above? Aren’t they too young to deal with a complex survey like this?

4. Please revise the phrase on line 156 into “…anterior nares swab samples.”

**Do you want your identity to be public for this peer review?** For information about this choice, including consent withdrawal, please see our Privacy Policy

Reviewer #1: No

Reviewer #3: No

---

## [Author Response · Author response to Decision Letter 2]

25 Mar 2025

Journal Requirements:

Authors: We confirmed that no references have been retracted and that the reference list is complete and correct.

Reviewers' comments:

Reviewer's Responses to Questions

Comments to the Author

Review Comments to the Author

Reviewer #1: (No Response)

Reviewer #3: 1. On page 6 lines 72-74, I suggest that the authors add an additional group of people other than those in university settings, for example, the global survivors of COVID-19. Thus, the sentence becomes like this: “Finally, surveillance for behavioral factors that mitigate or promote COVID-19 infection risk is important in developing public health recommendations targeting groups at high risk (17-20), including in university settings (21) and the global survivors of COVID-19 (36).” The 36th reference can be cited from a study by Fahriani et al. entitled “Persistence of long COVID symptoms in COVID-19 survivors worldwide and its potential pathogenesis -A systematic review and meta-analysis”.

Authors: Our references already include individuals in non-university settings across the globe. See:

18. Benham JL, Lang R, Kovacs Burns K, MacKean G, Leveille T, McCormack B, et al. Attitudes, current behaviours and barriers to public health measures that reduce COVID-19 transmission: A qualitative study to inform public health messaging. PLoS One. 2021;16(2):e0246941.

19. MacIntyre CR, Nguyen PY, Chughtai AA, Trent M, Gerber B, Steinhofel K, et al. Mask use, risk-mitigation behaviours and pandemic fatigue during the COVID-19 pandemic in five cities in Australia, the UK and USA: A cross-sectional survey. Int J Infect Dis. 2021;106:199-207.

In addition, the suggested reference does not seem relevant to the sentence as it does not address behavioral risk factors and public health recommendations. It is about long COVID symptoms and pathogenesis. Therefore, we have declined to include the reference.

2. This sentence on page 6 lines 74-76 will be deeper if authors could provide an example of the settings they mention before, such as healthcare professional settings. Thus the sentence will look like this “Integrating surveillance for active infection with surveillance for preventive behaviors can be successful in managing COVID-19 infection in some settings (22), such as the healthcare professional settings (37).” This 37th reference can be found in a study by Hamdan et al. entitled “Coping strategies used by healthcare professionals during COVID-19 pandemic in Dubai: A descriptive cross-sectional study”.

Authors: The suggested reference is not relevant for this sentence, as it does not discuss the integration of infection surveillance and preventive behavior surveillance. Therefore, we have declined to include the reference.

3. On page 9 table 1, the authors stated that one of the conditions that a participant was excluded from this study was being less than 2 years old. Does it mean that authors include minors starting from 3 years old and above? Aren’t they too young to deal with a complex survey like this?

Authors: We have clarified in sentence 122-123 that parents or guardians completed the survey for minors who could not complete it themselves, such as 2 year olds and 3 year olds.

4. Please revise the phrase on line 156 into “…anterior nares swab samples.”

Authors: This has been corrected.

---

## [Editor Report · Decision Letter 3]

13 Apr 2025

Prevalence of SARS-CoV-2 infection and immunity in a New York county in 2022 reveals frequent asymptomatic or undiagnosed infections

PONE-D-24-37725R3

Dear Dr. Cazer,

We’re pleased to inform you that your manuscript has been judged scientifically suitable for publication and will be formally accepted for publication once it meets all outstanding technical requirements.

Kind regards,

Harapan Harapan, MD, PhD

Academic Editor

PLOS ONE

---

## [Editor Report · Acceptance letter]

PONE-D-24-37725R3

PLOS ONE

Dear Dr. Cazer,

I'm pleased to inform you that your manuscript has been deemed suitable for publication in PLOS ONE. Congratulations! Your manuscript is now being handed over to our production team.

Kind regards,

on behalf of

Dr. Harapan Harapan

Academic Editor

PLOS ONE